# Characteristics and driving mechanisms of vegetation phenology variations in the Bosten Lake Basin, Xinjiang, China

**Kunyu Li[1], Xuemei Wang[1,2]\*, Zitai Liu[1], Cong Xue[1]**

**1** College of Geographic Science and Tourism, Xinjiang Normal University, Xinjiang, Urumqi, China,
**2** Xinjiang Uygur Autonomous Region Key Laboratory "Xinjiang Arid Lake Environment and Resources Laboratory", Xinjiang, Urumqi, China

\* xmwang2022@xjnu.edu.cn

## Abstract

Vegetation phenology functions as a highly sensitive biological metric for delineating the growth state of vegetation and mirroring environmental alterations. The exploration of its spatiotemporal variations and driving forces constitutes a key area within contemporary global change research. Leveraging MODIS NDVI data spanning from 2001 to 2023, this research derived phenology parameters for the Bosten Lake Basin in Xinjiang, China. The Sen+Mann-Kendall approach was utilized to analyze vegetation phenology trends, and the partial least squares path model (PLS-PM) was applied to quantitatively disclose the direct and indirect influences of climate, terrain, soil, and human activities on phenology. The findings indicated that over the past 23 years, the start of growing season (SOS) in the Bosten Lake Basin was predominantly concentrated between the 120th and 150th days, and it progressively advanced from the western to the eastern part. The end of growing season (EOS) fluctuated between the 240th and 270th days and gradually postponed from west to east. The length of the growing season (LOS) is mainly between 105 and 140 days and gradually lengthens from west to east. In the past 23 years, SOS advanced at an average rate of 8.9 days per decade, EOS advanced by 1.7 days per decade, and LOS extended by an average of 7.2 days per decade. The phenological parameters of diverse vegetation types demonstrated marked disparities. Needleleaf forests and cultivated vegetation exhibited earlier SOS, later EOS, and longer LOS, whereas alpine vegetation had the shortest growing season, approximately 73 days. With the increase of elevation, SOS was significantly delayed, EOS was significantly advanced, and LOS was significantly shortened ($p < 0.05$ for all). The direct impacts of climatic factors on SOS, EOS, and LOS were significant ($p < 0.01$), with total impacts of 1.03, −0.94, and −1.00, respectively. Among these climatic factors, temperature and precipitation are the most representative variables reflecting the influence of climate on phenology. Altitude indirectly affects phenology through climate,

**Data availability statement:** The data generated by the authors to support the results of this study is stored in a public repository. The data can be accessed at: https://github.com/Likunyu7/dataC.

**Funding:** This study was supported by the National Natural Science Foundation of China (Grant Nos. 42461042 and 41561051) and the Natural Science Foundation of Xinjiang Uygur Autonomous Region, China (Grant No. 2023D01A44). There was no additional external funding received for this study.

**Competing interests:** The authors declare no competing interests.

soil conditions, and human activities. Understanding the phenological change characteristics and driving mechanisms in the Bosten Lake Basin provides a basis for clarifying the ecological environment evolution and climate response in arid regions of China.

## Introduction

Vegetation, as a critical component of terrestrial ecosystems, plays a fundamental role in global energy balance and carbon cycling [1]. Vegetation phenology refers to the seasonal recurrence of plant life cycle events (e.g., budburst, leaf expansion, flowering, senescence), which are regulated by regional environmental factors such as temperature and precipitation [2,3]. As a comprehensive indicator of ecological responses to environmental changes, phenology reflects annual plant growth patterns and is among the most sensitive biological indicators of climate change [4,5]. With the accelerating impacts of global warming and anthropogenic activities, phenological research has gained significant attention in recent decades [6]. Traditional vegetation phenological observation relies on manual on-site recording of plant growth stages. This method provides accurate, species-specific data, and long-accumulated records aid regional studies. However, it has notable drawbacks: limited observation range, high time and labor costs, difficulty in large-scale synchronous monitoring, and weak data continuity and spatial representativeness due to human and environmental constraints. Advances in satellite remote sensing technology and data processing algorithms (such as the dynamic threshold and curve fitting methods) have enabled a shift from single-band, low-resolution to multi-spectral, high-spatiotemporal-resolution observations [7]. Remote sensing outperforms traditional methods by rapidly obtaining regional or global phenological data, overcoming spatial limitations. Satellite periodic observations facilitate long-term dynamic monitoring with reduced human interference, especially compensating for traditional method deficiencies in remote and harsh environments like deserts and plateaus [8].

In phenological monitoring through remote sensing, MODIS satellites offer a medium-spatial-resolution, long-term-series vegetation index dataset, which is suitable for large-scale phenological monitoring [9]. Among them, the normalized difference vegetation index (NDVI) is very sensitive to plant growth and can track the seasonal dynamics of vegetation. It is relatively simple to extract phenological parameters [8]. Key phenological parameters usually include the start of the growing season (SOS), the end of the growing season (EOS), and the length of the growing season (LOS). Many scholars have used MODIS NDVI time series data to conduct phenological studies. For example, researchers studied the phenological changes of vegetation in the arid area of northwestern Nigeria. The results showed that due to changes in precipitation patterns, shrubs and grasses in the region showed delayed greening, early senescence, and shortened growing seasons [10]. Similarly, some scholars analyzed the phenological changes of forests in Northeast China, indicating that rising temperatures led to an earlier

greening stage and a delayed senescence stage in forests [11]. In addition, scholars found that in the northern hemisphere, rising temperatures led to earlier SOS, later EOS, and longer LOS of vegetation [12,13]. The above studies confirm that MODIS NDVI is effective in accurately extracting vegetation phenological parameters, revealing plant growth patterns, and clarifying the impact of environmental changes on vegetation phenology. Vegetation phenology is influenced by complex interactions among climate, topography, soil properties, and human activities [14–16]. While climate (e.g., temperature, precipitation) exerts direct and indirect effects on plant development, spatial heterogeneity in terrain and soil characteristics introduces variability in phenological patterns [17]. Human activities, such as urbanization and agriculture, further modify phenological processes by altering local microclimates and land use [18]. Recent studies have attempted to disentangle the contributions of climate vs. anthropogenic factors using statistical methods (e.g., partial correlation analysis) [19,20]. However, traditional correlation-based methods often fail to account for the interaction and indirect effects of multiple variables, leading to potential bias in causal reasoning [21]. The Partial Least Squares Path Model (PLS-PM), a variance-based structural equation modeling (SEM) technique, addresses these limitations by quantifying direct and indirect relationships among variables while controlling confounding factors [22,23]. This method has been successfully applied to analyze phenological drivers in Central Asian arid regions and the Three Rivers Source region of China, providing mechanistic insights into multi-factor interactions [24,25]. By explicitly modeling causal pathways, PLS-PM offers a robust framework for dissecting the complex drivers of vegetation phenology.

The Bosten Lake Basin is located in the core of the arid zone in Central Asia and represents a typical inland lake basin in the arid region of northwest China. Its ecosystem is highly fragile, with vegetation phenology being exceptionally sensitive to climate change [26]. Under the combined effects of extreme aridity, fragile ecosystems, and human activity disturbances, this basin has become an ideal natural laboratory for studying the complex interactions between vegetation, climate, and human activities [27,28]. Investigating the spatiotemporal variation characteristics and driving mechanisms of vegetation phenology in the Bosten Lake Basin is crucial for revealing the response patterns of phenology to climate change and human activities. This study utilizes the MODIS NDVI dataset and employs the dynamic threshold method to extract the SOS, EOS, and LOS in the Bosten Lake Basin from 2001 to 2023, thereby profoundly clarifying the spatiotemporal differentiation rules of vegetation phenology. With the aid of the PLS-PM model, it quantifies the direct and indirect influences as well as the relative contributions of climatic factors, topography, soil properties, and human activities to vegetation phenology, while systematically elaborating on the driving mechanisms underlying phenological changes. This research seeks to achieve breakthroughs in three dimensions. At the data and methodological level, it will develop a research framework of "multi-parameter long-time series cooperative driving decoupling" applicable to inland lake basins in arid regions. In terms of mechanism analysis, it will precisely quantify the direct and indirect contributions of "nature-human activity" factors and clarify the response degrees of vegetation phenology in arid areas to complex stressors. In terms of application value, it will establish a quantitative linkage between the driving mechanism and ecological management, thereby providing direct and effective support for the ecological project of comprehensive management of the Bosten Lake Basin. Furthermore, this research will significantly advance the transformation of the research paradigm from "phenomenon description" to "mechanism analysis", offering a robust theoretical foundation and exemplary case support for the precise management of arid-region ecosystems.

## Materials and methods

### Research area

Located in the Bayingol Mongolian Autonomous Prefecture of Xinjiang, China (Fig 1), the Bosten Lake Basin lies on the northern edge of the Taklamakan Desert and the southern slope of the Tianshan Mountains. Named after Bosten Lake, the largest inland freshwater lake in China, the region's topography slopes from high in the northwest to low in the southeast. It has a temperate continental arid climate with long sunshine hours. Annual precipitation averages around 60 mm,

 

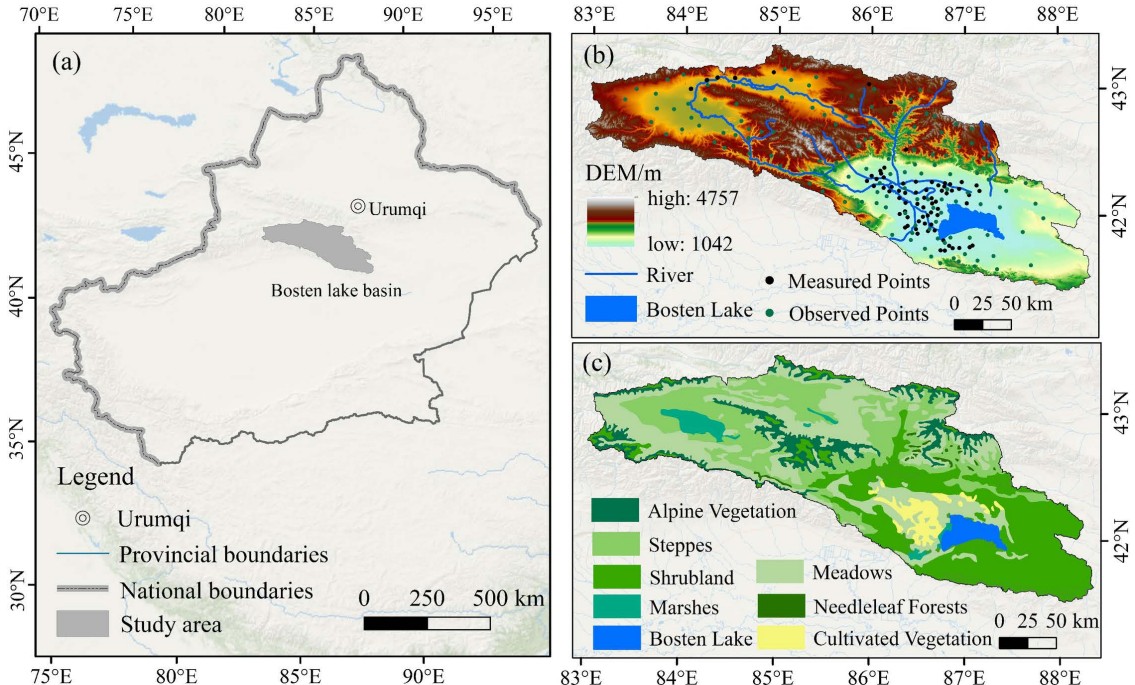

**Fig 1. Location of the study area, digital elevation model and vegetation distribution.** Note: The base map was created based on the standard map of the Ministry of Natural Resources of China (GS(2024)0650), with no modifications made to the boundaries of the base map.

with over 80% falling in summer. Basin rivers are primarily fed by melting snow and ice from high mountains, along with rainfall. In the basin, the surface landscape varies with elevation, transitioning from northwest to southeast through glacier snow zones, alpine meadows, oasis plains, desert grasslands, deserts, and finally, Bosten Lake. Vegetation types include Alpine Vegetation (AV), Meadows (Md), Marshes (Ms), Steppes (Sp), Needleleaf Forests (NF), Shrubland (Sl), and Cultivated Vegetation (CV). Significant ecological differences between the eastern and western basin, combined with human activities, have altered phenology. As a key water source protection area for "ecological water conveyance" in the lower Tarim River, the basin plays a vital ecological role [29]. The region follows a "mountain-oasis-desert" landscape pattern, with uneven spatial distribution and concentrated vegetation types. The striking elevation contrast between its eastern and western parts, along with diverse soils and notable variations in temperature and precipitation, makes phenology highly sensitive to environmental changes.

## Data sources and preprocessing

NDVI, defined as the ratio between the difference of near-infrared and red visible reflectance to their sum, is a widely used index for vegetation phenology analysis [30]. In this study, phenology data were obtained from two datasets. Data for 2001–2022 were sourced from MOD13Q1-NDVI, featuring a 16-day temporal resolution and a 250 m spatial resolution, while 2023's phenology was derived from NDVI values computed from the "sur_defl-b01" and "sur_defl-b02" bands of the MOD09A1 dataset, which have an 8-day temporal resolution and a 500 m spatial resolution. Both datasets were retrieved from the Google Earth Engine platform (https://earthengine.google.com/). To ensure research validity, the comprehensive NDVI dataset for 2001–2023 was standardized to a 16-day interval and a 250 m spatial resolution using MOD13Q1 and MOD09A1 data. Validation was carried out with field survey data collected in July and November 2023, comprising 72 measured and 140 observed sample points (Fig 1).

Vegetation growth is influenced by climate, terrain, and soil characteristics, resulting in significant spatial variability [31], while human activities further affect phenology to different extents. Consequently, this research selected four climatic (Clim) variables—temperature (Temp), precipitation (Pre), evapotranspiration (ET) and albedo (Albe); three terrain (Terr) factors—elevation (Elev), Slope, and Aspect; and two soil properties—soil type (SoilT) and soil moisture (SoilM). Additionally, three human activity (HA) variables—land use/land cover (LULC), population density (POP), and gross domestic product (GDP)—were identified as crucial drivers. Table 1 provides further details of all influencing factors. To facilitate subsequent data processing and analysis, all other data were resampled to 250-meter resolution using the nearest neighbor method, based on the spatial resolution of the primary data source (MODIS NDVI) used for phenology extraction. Additionally, all variables were standardized to the WGS-1984 coordinate system.

### Research methods

**Savitzky Golay filtering.** The vegetation index time series curve extracted from remote sensing data is often irregular and contains significant noise. Compared with methods such as median iterative filtering and Fourier transform, the time-series reconstruction method based on the Savitzky-Golay filter (S-G filter) can generate smoother and more visually coherent NDVI time series curves [32]. Subsequently, scholars have further improved this method and applied it to better fit vegetation growth curves [33,34]. The equation (1) is defined as follows:

$$j_*^Y = \sum_{i=-m}^{i=m} \frac{C_i Y_{j+i}}{N}$$

(1)

In the formula, $j_*^Y$ is the function fitting value, $Y_{j+i}$ is the original NDVI value, $N = 2m+1$ is the value of the function filtering window, and $C_i$ represents the weight of the $i$-th point.

**Dynamic threshold method.** The dynamic threshold method, also known as the local threshold method, was first proposed in 1997 [35]. Subsequently, this method was further improved [36]. Due to the varying intensity of changes in vegetation indices during different stages of vegetation growth, the dynamic threshold method was chosen to extract SOS and EOS values. This method effectively reveals the growth and development processes of vegetation. By considering

**Table 1. Influencing factors and data sources.**

| Data | index | Year | Spatial resolution | Data sources |
|---|---|---|---|---|
| Climate | Temp | 2001—2022 | 1km | China National Earth System Science Data Center (http://www.geodata.cn) |
| | Pre | | | |
| | ET | 2001-2023 | 500m | NASA (https://doi.org/10.5067/MODIS/MOD16A2.006) |
| | Albe | 2001-2023 | 500m | NASA (https://doi.org/10.5067/MODIS/MOD09GA.006) |
| Terrain | Elev | / | 30m | Google Earth Engine platform (https://earthengine.google.com/) |
| | Slope | / | | |
| | Aspect | / | | |
| Soil characteristics | SoilT | / | 250m | OpenLandMap (https://www.openlandmap.org/) |
| | SoilM | / | 1km | Digital Earth Open Platform (https://geovisearth.com/) |
| Human activity | LULC | 2020 | 30m | Zenodo (https://about.zenodo.org/) |
| | POP | 2000—2020 | 1km | Resource and Environment Science and Data Center of Chinese Academy of Sciences (http://www.resdc.cn/) |
| | | GDP | | |

the unique physiological characteristics in the study area, it deliberately excludes factors such as LULC and the intensity of human activities. Based on the actual conditions in the study area and relevant research findings of scholars [37], we establish a threshold for extracting the SOS at approximately 20% of the vegetation growth process. Similarly, using ground-observation data, we set a threshold for extracting the EOS at around 20% of the vegetation decay process. The calculation formula is as follows:

$$NDVI(SOS) = (NDVI_{max} - NDVI_{min1}) \times 20\%$$

(2)

$$NDVI(EOS) = (NDVI_{max} - NDVI_{min2}) \times 20\%$$

(3)

In equation (2), NDVI (SOS) represents the vegetation index threshold for extracting SOS, $NDVI_{max}$ represents the maximum vegetation index value, and $NDVI_{min1}$ represents the minimum vegetation index value during the upward phase of the NDVI curve; In equation (3), NDVI(EOS) represents the vegetation index threshold for extracting EOS, and $NDVI_{min2}$ represents the minimum vegetation index value during the descending stage of the NDVI curve. When the dynamic threshold reaches 20% of the difference between the maximum and minimum NDVI, SOS and EOS values are extracted. In this study, the extracted vegetation phenological parameters, namely SOS and EOS, were statistically analyzed using the Julian calendar [3].

**Sen+Mann Kendall trend analysis.** Using the Theil Sen Median estimation method and Mann Kendall trend test method to analyze the trend of vegetation phenological parameters over time. The Theil Sen Median estimation method has characteristics of high efficiency, insensitivity to measurement errors and outlier data, and is not affected by missing values and outliers, while the Mann Kendall test is suitable for trend significance testing of long time series data [38,39]. The calculating formula for trend values is as follows:

$$\beta = Median(\frac{x_j - x_i}{j - i}), 1 < i < j < n$$

(4)

In the formula, $x_i$ and $x_j$ are time series data; $\beta > 0$ indicates a delayed or extended trend, while $\beta < 0$ indicates an advanced or shortened trend. The statistical formula for Mann Kendall trend test is:

$$S = \sum_{i=1}^{n-1} \sum_{j=i+1}^{n} sgn(x_j - x_i)$$

(5)

$$sgn(x_j - x_i) = \begin{cases} 1, & x_j - x_i > 0 \\ 0, & x_j - x_i = 0 \\ -1, & x_j - x_i < 0 \end{cases}$$

(6)

In the formula, $S$ is the statistical measure for trend testing of time series data; $sgn$ is a sign function; $x_i$ and $x_j$ represent the values of the time series variable $x$ in the $i$-th and $j$-th periods, respectively; $n$ is the length of the time series data sample. When $n \geq 10$, the statistic $S$ approximately follows a standard normal distribution, and its standardized $Z$-value can be calculated by the following formula:

$$Z = \begin{cases} \dfrac{S-1}{\sqrt{Var(S)}}, & S > 0 \\ 0, & S = 0 \\ \dfrac{S+1}{\sqrt{Var(S)}}, & S < 0 \end{cases}$$

(7)

In the equation ([7]), Var (S) is the variance of the statistic S. Using a two-sided trend test, the critical value $Z_{(1-\alpha)/2}$ is found in a normal distribution table at a given significance level $\alpha$. When $|Z| \leq Z_{(1-\alpha)/2}$, the null hypothesis is accepted, indicating that the trend is not significant; If $|Z| > Z_{(1-\alpha)/2}$, the null hypothesis is rejected, and the time series S is considered to have a significant trend of change.

The trend values obtained through calculation were analyzed using Mann-Kendall's bilateral trend test, with confidence levels of 90%, 95%, and 99%. When $\beta > 0$ and $p < 0.01$, it indicates a highly significant delay (extension); when $0.01 < p < 0.05$, it indicates a significant delay (extension); and when $0.05 < p < 0.1$, it indicates an insignificant delay (extension). When $\beta < 0$ and $p < 0.01$, it indicates a highly significant advance (reduction); when $0.01 < p < 0.05$, it indicates a significant advance (reduction); and when $0.05 < p < 0.1$, it indicates an insignificant advance (reduction). When $\beta = 0$, there is no change.

**Introduction to the PLS-PM model.** Partial Least Squares Path Modeling (PLS-PM) is a correlation-based structural equation modeling approach that is particularly effective for handling small sample sizes and non-normal data [22]. In this method, causality is conveyed through linear conditional expectations, aiming to identify the most accurate linear predictive relationships while leveraging latent variables to model complex causal interactions or prediction models. PLS-PM is generally divided into two components: an external model that associates observed indicators with their latent constructs, and an internal model that links various latent variables. Within this framework, path coefficients measure both the strength and direction of direct effects between variables—a positive coefficient signifies that increases in an explanatory variable are accompanied by increases in the response variable, and vice versa. A Goodness of Fit (GOF) index, constructed via global measurement techniques, is used to evaluate the model's quality, with values exceeding 0.36 suggesting reliable outcomes [40]. Before performing the path analysis, all data are normalized. In this study, the observed variables include SOS, EOS, and LOS, while the latent variables encompass factors related to climate (Clim), terrain (Terr), soil, and human activities (HA). Specifically, the Clim factors are Temp, Pre, ET and Albe; the Terr factors are Elev, Slope, and Aspect; the soil factors are SoilT and SoilM; and the HA factors are LULC, POP, and GDP.

## Results and analysis

### Verification of vegetation phenological extraction results

To enhance the precision of extracting phenological parameters from remote sensing data, the research team carried out field investigations for phenological validation in March, July, and November 2023. In total, 72 field sample points were gathered for phenological survey data to validate the results obtained from remote sensing data. Moreover, referring to the research of scholars [41] on phenology in Xinjiang, China, 140 randomly selected sample points were used as observed values to further verify the extraction results of this study. The research results are presented in Fig 2. Analysis shows that the Start of Season (SOS) extracted in this study has a good match with both measured and observed values, with the coefficient of determination ($R^2$) being 0.90 ($p < 0.05$) and 0.93 ($p < 0.05$) respectively. Meanwhile, the coefficient of determination for the End of Season (EOS) is 0.89 ($p < 0.05$) and 0.91 ($p < 0.05$). The validation results prove that the phenology parameters extracted using remote sensing data and dynamic threshold methods are highly consistent with ground-measured results, suggesting that remote sensing for monitoring phenology is highly accurate and reliable.

### Interannual variation of vegetation phenology

Based on the MODIS NDVI dataset, the vegetation phenological parameters (SOS, EOS, and LOS) of the Bosten Lake Basin were extracted. The multi-year average value of these phenological parameters was calculated to obtain the inter-annual variation results of vegetation phenology in the study area from 2001 to 2023 (as shown in Fig 3). As can be seen from the figure, over the past 23 years, the start time of the vegetation growth season in the Bosten Lake Basin has been significantly advanced ($R^2 = 0.26$, $p < 0.05$), with an average advance rate of approximately 8.9 days per decade. The start

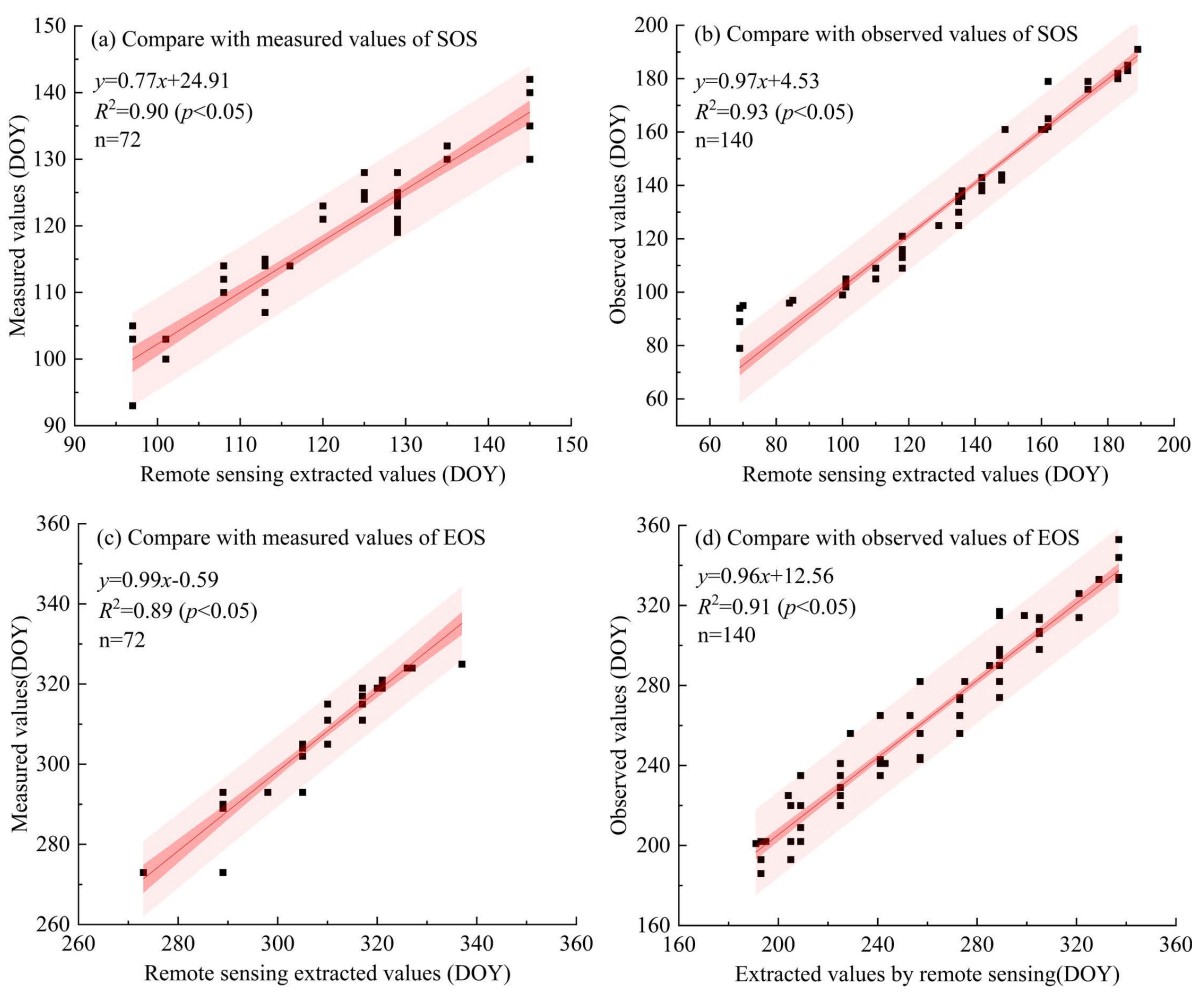

**Fig 2. Accuracy evaluation of vegetation phenology extraction.**

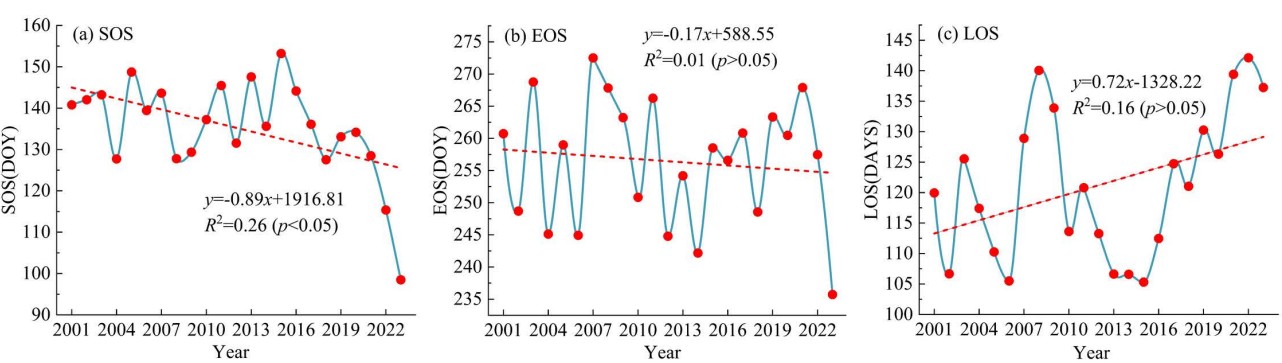

**Fig 3. Interannual Variability of Vegetation Phenology in the Bosten Lake Basin from 2001 to 2023.**

time of the vegetation growth season is mainly concentrated within the range of the 120th to the 150th day, and in 2023, the start time of the vegetation growth season is the earliest, about the 98th day. However, the end time of the growth season shows no significant advancing trend ($R^2 = 0.01$, $p > 0.05$), with an average advance rate of about 1.7 days per decade. The end time of the vegetation growth season is mainly concentrated within the range of the 240th to the 270th day, and in 2023, the end time of the growth season is the earliest, about the 236th day. The length of the vegetation growth season generally shows an extension trend ($R^2 = 0.16$, $p > 0.05$), and the average extension rate is about 7.2 days per decade. The length of the growing season is mainly between 105 and 140 days. The length of the vegetation growing season in 2022 is the longest, about 142 days, while the growth season in 2015 is the shortest, about 105 days. In general, the vegetation phenological parameters have fluctuated over the past 23 years. The start and end of the growing season are advanced, and the length of the growing season shows an increasing trend.

Through further analysis of the phenological changes of different vegetation types in the Bosten Lake Basin, the average phenological periods over many years are depicted in Fig 4. Among them, the growth season of NF commences the earliest, approximately on the 111th day. Conversely, the growth season of AV starts at the latest, and the vegetation does not turn green until the 162nd day. Simultaneously, the start time of the growth season of CV and Sl is significantly earlier than that of Sp, Ms, and Md. The end time of the growth season of different vegetation types exhibits a different trend. The end time of the growth season of AV is the earliest, approximately on the 237th day. Whereas the end time of the growth season of CV is the latest, around the 278th day. Additionally, the length of the growth season of different vegetation types is closely related to the beginning and end of the growth season. As can be observed from the figure, the growth season of NF and CV begins earlier and ends later; hence their growth season is longer. The length of the growing season of NF is about 166 days, while that of CV is approximately 161 days. On the contrary, the growth season of AV and Md starts late and ends earlier. The length of their growth season is less than 100 days. The length of the growth season of AV is the shortest, about 73 days, and that of Md is around 97 days. The analysis demonstrates that vegetation phenology in the Bosten Lake basin displayed significant spatial variability from 2001 to 2023, driven by the interplay of vegetation types, regional climatic conditions, and hydrological processes.

Overall, the forest ecosystem dominated by NF and the artificial oasis vegetation composed of CV exhibit a relatively long growing season. The grassland ecosystem, consisting of AV and Md, is situated at the middle and high altitudes of the basin, with a relatively short LOS. The wetland ecosystem dominated by Ms is primarily distributed in Bayanbulak Wetland and on the west bank of Bosten Lake. Vegetation in the high-altitude Bayanbulak Wetland has a late SOS, an early EOS, and a relatively short LOS. In contrast, low-altitude wetland vegetation on the west bank of Bosten Lake exhibits an earlier SOS and a later EOS. These results indicate that the phenology of Ms displays significant spatial differences.

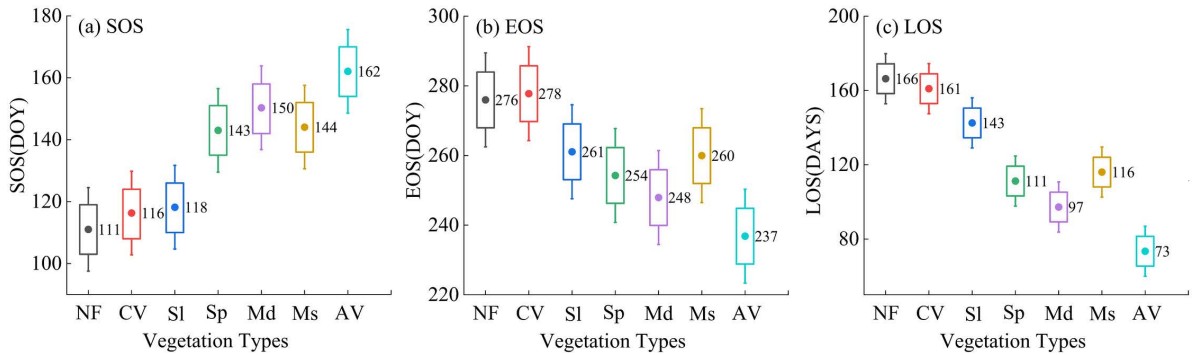

**Fig 4. Annual average phenological parameters of different vegetation types from 2001 to 2023.** Note: AV, Md, Ms, Sp, NF, Sl, and CV represent Alpine Vegetation, Meadows, Marshes, Steppes, Needleleaf Forests, Shrubland, and Cultivated Vegetation, respectively.

## Spatial variation of vegetation phenology

Fig 5 illustrates the spatial distribution and statistical outcomes of the multi-year average value of phenology in the Bosten Lake Basin from 2001 to 2023. As depicted in Fig 5 (a) and Fig 5 (b), the SOS in the basin shows a gradual delay from east to west. It varies between the 85th and 185th days, with an average SOS of approximately the 129th day. There are two peaks in the spatial pixel proportion of SOS. The first peak occurs around the 120th day, mainly distributed in the middle and east of the basin, while the second peak takes place around the 165th day, mainly in the western part of the basin.

From Fig 5 (c) and Fig 5 (d), it can be observed that the EOS in the basin advances gradually from east to west. The EOS changes occur around the 200th to 320th day, being concentrated between the 240th and 290th days and reaching

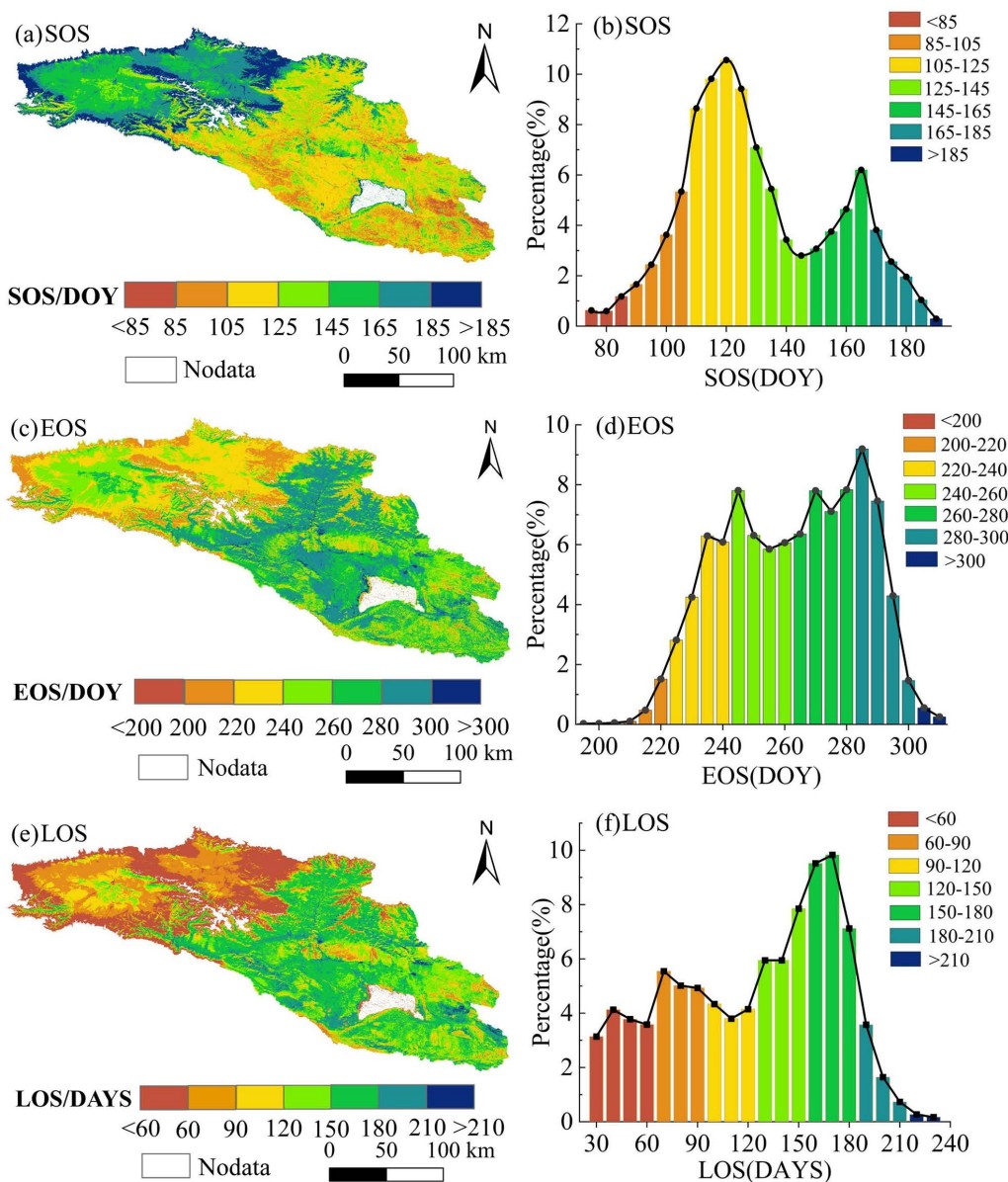

**Fig 5. spatial variation and area proportion of vegetation phenology from 2001 to 2023.**

a peak on the 285th day. The average EOS is about the 261st day. The EOS before the 240th day is primarily distributed in the high-elevation mountainous regions in the west and north of the basin, and the EOS after the 240th day is mainly concentrated in the oasis area in the middle and east of the basin. The LOS is mainly determined by the SOS and EOS. As shown in Fig 5 (e) and Fig 5 (f), the LOS of vegetation gradually shortens from east to west, and the spatial difference is distinct. The LOS is mainly concentrated within the range of 60–210 days, with an average length of 132 days. The peak values of LOS occur around 65 days and around 170 days. The LOS around 65 days is mainly distributed in the high-elevation mountains in the west of the basin, and the LOS around 170 days is mainly distributed in the oasis area with relatively low elevation in the east of the basin. Due to the influence of factors such as climate, terrain, soil, and human activities, the water and heat conditions in the eastern and western regions of the basin are quite different, and the vegetation types also vary. As a result, the spatial variation in phenology displays an obvious zonal distribution pattern.

To investigate the relationship between vegetation phenology and elevation, a total of 3,811 sampling points were randomly selected (with intervals exceeding 1 km), ensuring an even distribution across various elevation gradients. The corresponding values for vegetation phenological parameters—specifically SOS, EOS and LOS—were extracted for each point. Scatter plots were then generated with elevation on the horizontal axis and the respective vegetation phenological parameters on the vertical axis. A trend line was fitted to illustrate how phenology varies with changes in elevation (Fig 6). The analysis revealed that the determination coefficients of the regression equations for SOS, EOS, and LOS in relation to elevation ranged from 0.118 to 0.312, indicating a significant correlation ($p < 0.05$) among various vegetation phenological parameters and elevation. Notably, as elevation increases, there is a discernible delay in the start time of the growing season; specifically, this delay occurs at a rate of approximately 0.020 days per meter increase in elevation. Consequently, when elevation rises by 100 meters, SOS is delayed by roughly two days. Furthermore, the slopes of the regression equations for EOS and LOS relative to elevation are −0.013 and −0.026 respectively; these results suggest that the end time of the growing season advances while its duration shortens as elevations increase. For every additional 100 meters in elevation gained, EOS tends to advance by about 1.3 days while LOS decreases by approximately 2.6 days. The above results indicate that elevation has a significant impact on phenology. As elevation increases, the start time of the growing season is delayed, the end time of the growing season is advanced, and the length of the growing season tends to shorten.

## Trend analysis of vegetation phenology

The changing trends of vegetation phenology in the Bosten Lake Basin from 2001 to 2023 were analyzed (as shown in Fig 7). The areas where the SOS and EOS advanced in the basin were larger than those of the areas with delays. Specifically, areas where the SOS and EOS advanced at a rate of less than 0.5 days per year accounted for approximately 73.08% and 70.19%, respectively. Areas with an SOS advance of 1 day or more accounted for 20.05%, mainly distributed in the shrubs around the artificial oasis. However, areas with an EOS advance of more than 1 day accounted for only

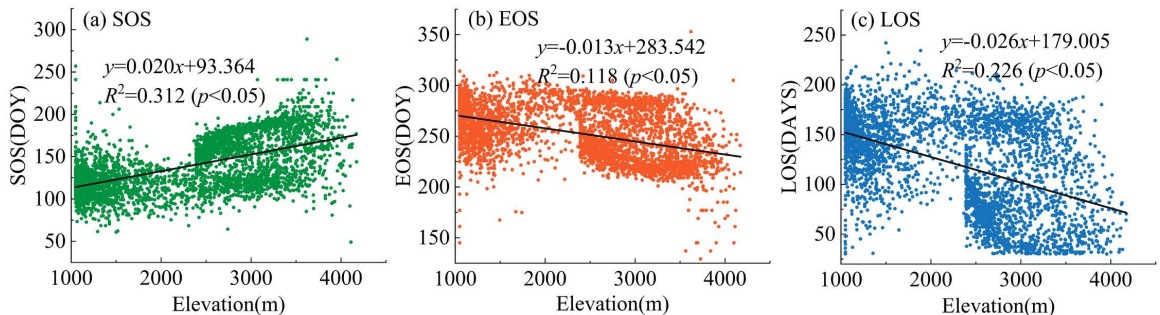

**Fig 6. Changes of vegetation phenology at different elevation gradients from 2001 to 2023.**

   

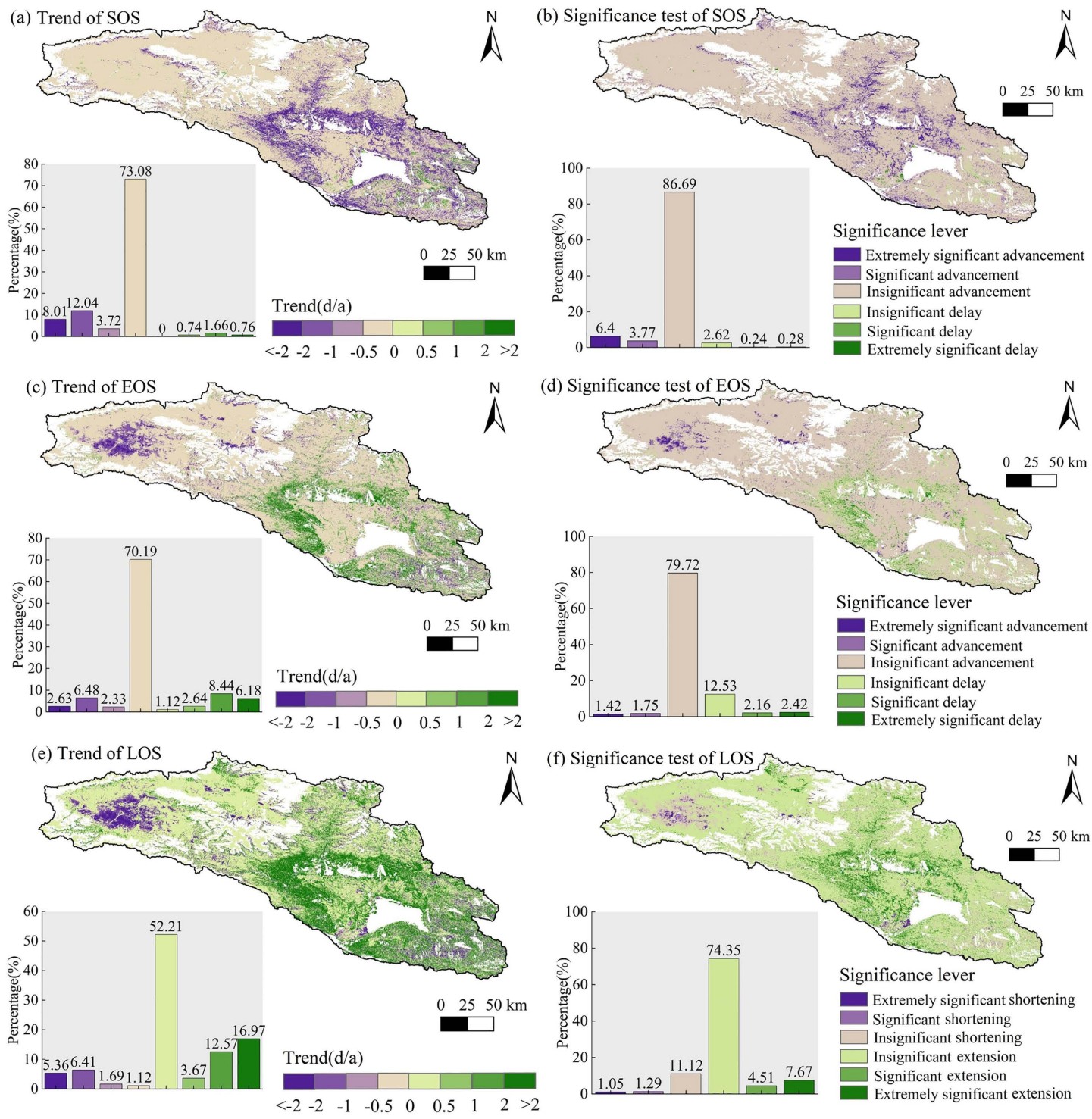

**Fig 7. Change trend and significance analysis of vegetation phenology from 2001 to 2023.**

9.11%, mainly distributed in the Bayinbuluk wetland. Among them, areas with an extremely significant ($p<0.01$) SOS and EOS advance accounted for 6.40% and 1.42%, respectively. The LOS showed an overall extension trend across the entire basin. About 55.88% of the areas had an annual extension of less than 1 day, and 74.35% of the areas showed no significant extension trend. Only 12.18% of the areas exhibited significant ($p<0.05$) or extremely significant ($p<0.01$) LOS extension, mainly located in shrub and cultivated areas. Additionally, the proportions of areas with significantly ($p<0.05$) shortened and extremely significantly ($p<0.01$) shortened LOS were 1.29% and 1.05%, respectively, mainly in marshes. Overall, the SOS and EOS in the Bosten Lake Basin showed an advanced trend, and the LOS showed an extension trend.

By analyzing the phenological change trends of various vegetation types (Fig 8), it is evident that the SOS for all seven vegetation types is advancing. SI shows the most pronounced advancement, with an average increase of approximately 0.8 days per year, while Ms exhibits the least advancement in SOS. Concurrently, the EOS for four vegetation types is advancing, while the EOS of the other three types is experiencing a delay. Except for Ms, the LOS for the remaining six vegetation types shows a trend of prolongation. Notably, SI has the greatest extension in its growing season, averaging an increase of 1.43 days per year, followed by CV with an extension of 0.76 days per year. The trends and variations in SOS and EOS directly affect whether LOS shortens or lengthens. The SOS and EOS of NF, Sp, and AV show an advancing trend. Notably, the advancement of SOS is more significant than that of EOS, resulting in an overall extended LOS for these categories. Similarly, for CV, SI, and Md, the SOS shows advancements while the EOS shows delays. Here, too, the magnitude of the SOS advancement exceeds that of the EOS delay, leading to a prolonged LOS. In contrast to these patterns, both the SOS and EOS of Ms show advancements. However, it should be noted that the amplitude of the SOS advance is less than that of the EOS advance. Therefore, it results in a shortening trend in its LOS.

## Analysis of factors affecting vegetation phenology

Fig 9 illustrates the correlation analysis of phenological parameters and their influencing factors. Evidently, there is a highly significant correlation between climatic factors and phenology ($p<0.01$). Specifically, Tem exhibits a negative correlation with the SOS, whereas Pre shows a positive correlation with SOS. This indicates that the higher the temperature, the earlier the vegetation rejuvenation period, and the higher the precipitation, the later the vegetation rejuvenation period. Tem is positively correlated with both EOS and LOS. By contrast, Pre is negatively correlated with EOS and LOS. That

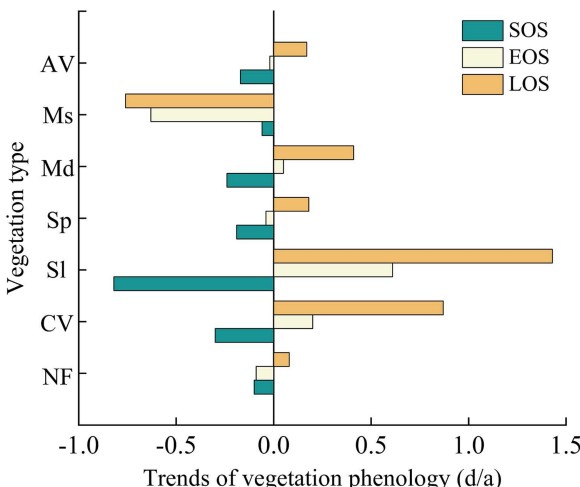

**Fig 8. The phenological change trend of different vegetation type from 2001 to 2023.**

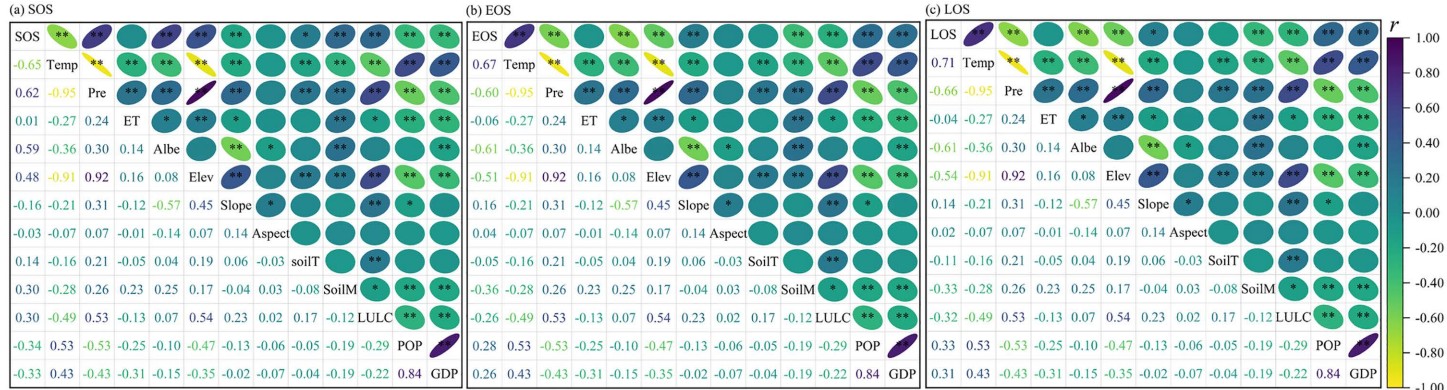

**Fig 9. Results of correlation analysis between phenology and influencing factors.** Note: * and ** represent p<0.05 and p<0.01, respectively. Temp, Pre, ET, Albe, Elev, SoilT, SoilM, LULC, POP, and GDP represent temperature, precipitation, evapotranspiration, albedo, elevation, soil type, soil moisture, land use/land cover, population density and gross domestic product, respectively.

is, higher temperatures result in a later EOS for vegetation and a longer LOS. Simultaneously, more precipitation causes an earlier EOS for vegetation and a shorter LOS. The correlation between phenology and ET is relatively weak, with a correlation coefficient below 0.1. In contrast, the correlation with Albe is significantly higher (p<0.01), and the correlations with Temp and Pre are comparable. Specifically, SOS is positively correlated with Albe, whereas EOS and LOS exhibit negative correlations with Albe. Regarding terrain factors, the correlation between elevation and phenology is significant (p<0.01). However, the correlations between slope, aspect, and phenology are not significant. This is mainly due to the substantial elevation disparity between the east and west of the basin. This disparity impacts the temperature and precipitation in specific areas, thereby influencing the growth of vegetation. Among soil-related factors, the correlation between soil type and phenology is the most prominent. Due to the differences in elevation and climate conditions, the environmental factors for soil development vary, giving rise to a wide array of soil types. These diverse soil types, in turn, promote the growth of different vegetation types, causing the phenology to display distinct spatial variances. Factors associated with human activities, such as LULC, POP, and GDP, show a low correlation with phenology, with correlation coefficients less than 0.3. Thus, it can be deduced that natural environmental factors are the primary determinants of the changes in phenology characteristics in the Bosten Lake Basin. In comparison, the impact of human activities on phenology is relatively feeble.

## Path analysis of influencing factors on vegetation phenology

To further uncover the driving path of vegetation phenology in the Bosten Lake Basin, the PLS-PM method was utilized to analyze the direct and indirect effects of various influencing factors on phenology parameters. As depicted in Fig 10, the goodness of fit index (GOF) of the three phenological parameters all exceeds 0.45. This indicates that the model exhibits good overall adaptability to the data and can more precisely elucidate the relationships among climate, terrain, soil, human activities, and their impacts on the SOS, EOS, and LOS. From Fig 10 (a), the maximum direct impact of climate on SOS is 0.97, demonstrating a highly significant positive effect (p<0.001). The indirect effect of climate factors on SOS via soil and human activities is 0.06. Considering both the direct and indirect effects, the total effect is 1.03. The direct effect of topography on SOS is −0.40, and the indirect effect of topography on SOS through climate, soils, and human activities is 0.80, for a total effect of 0.40. The direct effect of soil on SOS was 0.08, the indirect effect tended to be 0, and the total effect was 0.08. The direct effect of human activities on SOS is merely −0.03. These results suggest that climate is the primary factor influencing the change of SOS, and human activities have the least impact. As shown in Fig 10 (b), the direct

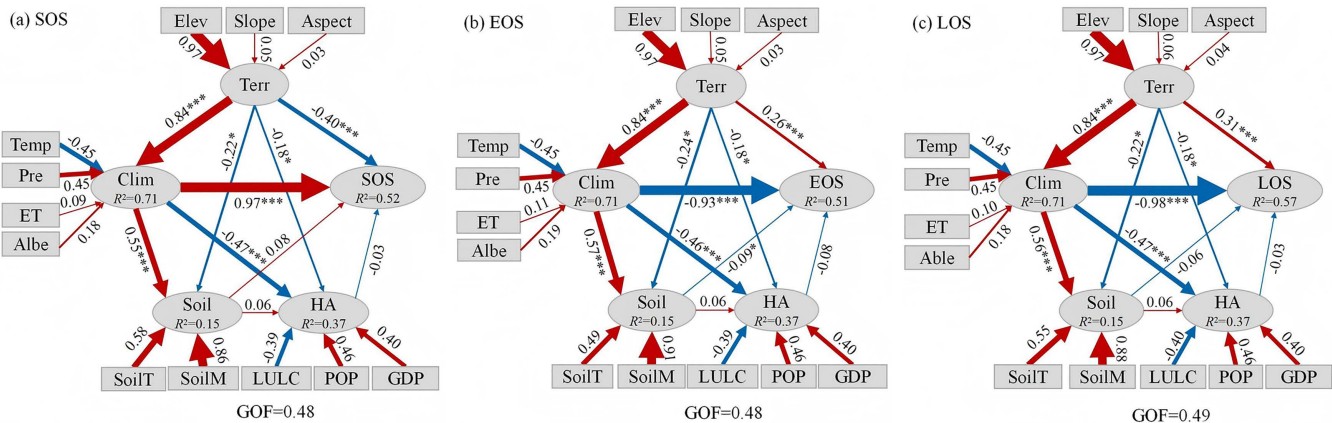

**Fig 10. The influence of the path of vegetation phenology.** Note: Red arrows indicate positive effects, blue arrows indicate negative effects, and the numbers next to the arrows are path coefficients, which indicate the magnitude of the relationship's effect. *, **, and *** represent, p < 0.05, p < 0.01, and p < 0.001, respectively. R2 indicates the proportion of the total variation in the dependent variable that is explained by the independent variable.

impacts of climate, terrain, soil, and human activities on EOS are −0.93, 0.26, −0.09, and −0.08, respectively. The indirect effect of climate on EOS through soil and human activities is −0.01. Considering its direct and indirect effects, the total effect of climate on EOS is −0.94. The indirect effect of terrain factors on EOS through climate, soil, and human activities is −0.75, and the total effect is −0.49. The indirect effect of soil on EOS tends to be 0. Combining with its direct effect, the total effect of soil on EOS is −0.09. The direct effect of human activities on EOS is small at −0.08 and insignificant. In conclusion, Climate, topography, soil and human activities all have a negative effect on SOS. Climate has the greatest effect on EOS and human activities have the least effect on EOS. As presented in Fig 10 (c), the direct impact of climate on LOS is −0.98, and the impact is extremely significant (p < 0.001). The indirect effect of climate on LOS through soil and human activities is −0.02. Considering the direct and indirect effects, the total effect of climate on LOS is −1.00. The indirect effect of terrain on LOS through climate, soil, and human activities is −0.80. Considering the direct and indirect effects, the total effect of terrain on LOS is −0.49. The total effect of soil on LOS is −0.06. It can be observed that climate has the greatest impact on LOS, and human activities have the least impact on LOS. To summarize, the direct effect of climate on the SOS, EOS, and LOS of vegetation in the Bosten Lake Basin is the most conspicuous, and temperature and precipitation is the variable that can best reflect the impact of climate on phenology. The effects of evapotranspiration and albedo on vegetation phenology were smaller than those of temperature and precipitation. Among terrain factors, elevation is the most crucial influencing variable, followed by slope and aspect. POP is the optimal indicator of human activities. Additionally, Soil moisture is a good indicator of soil characteristics and has a strong influence on the phenological period.

## Discussion

### Uncertainty analysis of vegetation phenology extraction

The results show that in the past 23 years, both SOS and EOS in the Bosten Lake Basin have shown an advance trend, while LOS has shown an extended trend. Specifically, SOS usually occurs between the 120th and 150th days, while EOS mainly occurs between the 240th and 270th days, and LOS ranges from 105 days to 140 days. SOS is significantly advanced (p < 0.05), EOS is not significantly advanced, and LOS is not significantly extended. With the increase in altitude, the start time of the vegetation growing season in mountainous areas is later than that in plain areas. In particular, the greening period of high-altitude grasslands and meadows in the western mountainous areas is significantly later than that of shrubs and cultivated plants in the eastern plains. There are obvious vertical

differentiation patterns and zonal differences in the greening period of different vegetation types. In general, the phenological characteristics of the Bosten Lake Basin are that SOS and EOS are in an advance trend, and LOS is in an extended trend. This result is consistent with the previous research results on the phenology of vegetation in the mountainous areas of Xinjiang [37,42]. In addition, research on grassland phenology in Xinjiang shows that the greening period in the southern Tianshan Mountains, temperate grasslands, alpine grasslands, and meadows also shows a trend of starting earlier, which verifies the reliability of our research results [43,44]. At the same time, the study further showed that the greening period of vegetation in the Tianshan region of Xinjiang was significantly advanced, which may be related to the early onset of SOS caused by rising temperatures [45]. The phenological extraction results of this study also verified this view. Therefore, combining remote sensing technology, phenological models, and ground observations to establish a "space-air-ground" multi-scale integrated monitoring system is the key to achieving accurate monitoring of vegetation phenology [9,46].

Jiang et al. [47] found that the start of the SOS in the mid-high latitude regions of the Northern Hemisphere predominantly occurs within the 140th-160th day range. Since 2009, the SOS has exhibited an advancing trend at a rate of 1.3 days per decade. This study corroborates this conclusion; however, our research reveals that the advancing rate of SOS in the Bosten Lake Basin is significantly higher, suggesting that the phenology in this basin is highly sensitive to climate warming. This finding aligns with the research conclusion of Yin et al. [48]. Moreover, this study discovered that the altitude gradient has an amplifying effect on the delay of SOS and the advancement of EOS ($p<0.05$). Gao et al. [49] also indicates that phenology displays certain spatial heterogeneity across different altitude gradients. Consequently, phenology shows distinct spatial differences among regions, and these differences are especially pronounced in arid areas. This phenomenon not only reflects the uniqueness of the hydrothermal coupling drive in arid regions but also highlights the indicative role of vegetation phenology in climate change.

## Driving factor analysis based on the PLS-PM method

Using the PLS-PM method, we quantitatively examined how climate, terrain, soil, and human activities influence phenology in the Bosten Lake Basin. The analysis revealed that climate and terrain factors exhibit the strongest correlations with phenology, with temperature and elevation having a particularly pronounced effect, consistent with previous findings [50]. Specifically, the SOS was negatively related to temperature and positively related to precipitation. Within an optimal temperature range, higher temperatures at the onset enhanced plant physiological activities and accelerated growth, leading to an earlier start of the growing period. Conversely, EOS and LOS showed a positive correlation with temperature and a negative correlation with precipitation; higher temperatures promoted photosynthesis and nutrient uptake, thereby delaying phenological end and extending the growing season, whereas increased precipitation could result in excessive soil moisture that hampers root respiration and nutrient absorption, ultimately reducing the growing season [27].

In the Bosten Lake Basin, surface albedo significantly affects vegetation phenology, with a close relationship between the two. Surface albedo directly impacts plants' growth environment and phenological processes by altering the surface water and thermal regulation [51,52]. In spring, when vegetation sprouts, surface albedo decreases due to increased vegetation coverage. In autumn, after vegetation turns yellow, reduced vegetation coverage leads to more exposed soil, and surface albedo increases accordingly. In correlation analysis, the correlation between albedo and phenology is comparable to that between temperature, precipitation and phenology, and much stronger than that between evapotranspiration and phenology. However, in PLS-PM analysis, the influence of albedo on phenology is weaker than that of temperature and precipitation. This discrepancy arises from differences in calculation methods: correlation analysis only assesses the relationship between two variables, while the PLS-PM method quantifies both direct and indirect influences between variables, thus enabling more accurate estimation of the overall influence of each factor on phenology [53–55]. This study offers a novel perspective by analyzing the interactions among various driving factors in the Bosten

Lake Basin, thus compensating for the lack of quantitative analyses in previous research. However, the PLS-PM method's assumption of linear relationships between latent variables limits its ability to capture nonlinear interactions. Future research could integrate machine learning and other analytical techniques to address these limitations and enhance the accuracy of factor analysis [56].

## Limitation

Due to variations in the spatiotemporal resolutions of remote sensing data, the extracted phenological characteristics may contain certain uncertainties. The MOD13Q1 data used in this study is preprocessed NDVI product data, with a spatial resolution of 250m and a temporal resolution of 16 days. It boasts high data quality and stability, making it suitable for time series analysis. MOD09A1 provides surface albedo data with a spatial resolution of 500 meters and an 8-day temporal resolution, offering high temporal flexibility. However, NDVI derived from this product's albedo data requires manual calculation, and issues such as cloud occlusion and atmospheric correction also need to be addressed. This increases the complexity of data processing, and preprocessing methods may affect the results, leading to certain estimation errors compared with vegetation phenological data extracted from MOD13Q1 NDVI data [57]. To ensure result accuracy, this study combined the two types of data and verified the findings using ground observation data, thereby enhancing the reliability of phenological extraction. Additionally, this study only analyzed the influence of climatic factors such as annual average temperature and precipitation on phenology. In future research, the impacts of pre-season factors such as maximum temperature, minimum temperature, and precipitation on phenology could be examined in greater detail [58]. Meanwhile, The coupling effect of driving factors increases uncertainty in exploring phenological driving mechanisms. Specifically, the interactions of climatic factors (Tem, Pre, et al.) vary by region. Meanwhile, the superimposed influences of human activities (e.g., LULC, irrigation) and natural factors are difficult to precisely disentangle. Additionally, phenology's response to environmental factors exhibits a certain temporal effect. Therefore, future research should integrate multi-source data fusion and multi-scale model coupling to reduce uncertainties in driving mechanism analysis [50].

## Conclusion

In this study, the change trends of vegetation phenology in the Bosten Lake Basin, China from 2001 to 2023 were analyzed, and the impacts of climate, terrain, soil, and human activities on phenology were discussed. The findings are as follows: (1) Over the past 23 years, the SOS of vegetation in the Bosten Lake Basin has advanced at an average rate of 8.9 days per decade. The EOS has also shown a slightly advancing trend, with an average advancement of 1.7 days per decade. Correspondingly, the LOS has extended by an average of 7.2 days per decade. (2) Spatially, the SOS and EOS gradually show a delaying trend from east to west, while the LOS gradually shortens from east to west. Cultivated vegetation in the eastern part has an earlier start and a later end of the growing season, resulting in a significantly longer growing-season length. In contrast, alpine vegetation in the western part has the latest start and the earliest end of the growing season, with the shortest growing season. With the increase in elevation, the SOS shows a delaying trend, while the EOS and LOS show slightly advancing and shortening trends respectively. (3) Climate exerts a significant impact on the SOS, EOS, and LOS of vegetation in the Bosten Lake Basin, with total influence coefficients of 1.03, −0.94, and −1.00, respectively. SOS is negatively correlated with Tem but positively correlated with Pre and Albe. Conversely, EOS and LOS are positively correlated with Tem and negatively correlated with Pre and Albe. Both climate and Elev significantly influence phenology, while soilM and human activities also contribute to certain impacts on phenology.

## Acknowledgments

We thank the Xinjiang Laboratory of Lake Environment and Resources in Arid Zone, China for technical support. We also thank the reviewers for their valuable comments that improved the quality of this paper.

## Author contributions

**Conceptualization:** Kunyu Li, Zitai Liu.

**Data curation:** Cong Xue.

**Funding acquisition:** Xuemei Wang.

**Methodology:** Kunyu Li, Zitai Liu.

**Software:** Kunyu Li, Zitai Liu.

**Supervision:** Xuemei Wang, Cong Xue.

**Writing – original draft:** Kunyu Li, Zitai Liu.

**Writing – review & editing:** Xuemei Wang.

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
