## [Decision Letter · Decision Letter 0]

14 May 2025

PONE-D-25-17547Characteristics and Driving Mechanisms of Vegetation Phenology Variations in the Bosten Lake Basin, Xinjiang, ChinaPLOS ONE

Dear Dr. Wang,

Thank you for submitting your manuscript to PLOS ONE. After careful consideration, we feel that it has merit but does not fully meet PLOS ONE’s publication criteria as it currently stands. Therefore, we invite you to submit a revised version of the manuscript that addresses the points raised during the review process.

We look forward to receiving your revised manuscript.

Kind regards,

Dafeng Hui, Ph.D.

Academic Editor

PLOS ONE

 [This study was supported by the National Natural Science Foundation of China (Grant No. 42461042; 41561051) and the Natural Science Foundation of Xinjiang Uygur Autonomous Region, China (Grant No. 2023D01A44).].

6. We note that Figures 1a-c, 5a,5c,5e, 7a-f in your submission contain [map/satellite] images which may be copyrighted. All PLOS content is published under the Creative Commons Attribution License (CC BY 4.0), which means that the manuscript, images, and Supporting Information files will be freely available online, and any third party is permitted to access, download, copy, distribute, and use these materials in any way, even commercially, with proper attribution. For these reasons, we cannot publish previously copyrighted maps or satellite images created using proprietary data, such as Google software (Google Maps, Street View, and Earth). For more information, see our copyright guidelines: http://journals.plos.org/plosone/s/licenses-and-copyright.

1. You may seek permission from the original copyright holder of Figures 1a-c, 5a,5c,5e, 7a-f to publish the content specifically under the CC BY 4.0 license. 

Additional Editor Comments:

I now received one report from an expert review who consider the study has merits but also raised some technique concerns. The authors need to make a substantial revision before it can be recommended for publication.

Reviewers' comments:

Reviewer's Responses to Questions

**Comments to the Author**

1. Is the manuscript technically sound, and do the data support the conclusions?

Reviewer #1: Yes

2. Has the statistical analysis been performed appropriately and rigorously? 

Reviewer #1: Yes

3. Have the authors made all data underlying the findings in their manuscript fully available?

Reviewer #1: Yes

4. Is the manuscript presented in an intelligible fashion and written in standard English?

Reviewer #1: Yes

5. Review Comments to the Author

Reviewer #1: This study based on MODIS NDVI data to investigated the response of vegetation phenology to climate change and human activities in the Bosten Lake Basin, Xinjiang, China from 2001 to 2023. The results of this study may contribute to the exploration of environmental protection and sustainable development in response to climate change in Northwest China. However, there are some concerns that the authors should address before it can be considered for publication.

1. More mechanism explanations should be added to further explain the impact of climate change and human activities on vegetation phenology.

2. In the last paragraph of the introduction, the authors should highlight the significance of this study.

3. In Figure 1, the compass is not aligned with the direction of the latitude/longitude grid. I suggest the authors delete the compass and add the label of “China” in the Figure 1(a).

4. How did the authors account for inconsistent spatial resolution between different data sources?

5. How did authors consider the impact of other environmental variables (e.g., albedo, evapotranspiration, etc.) on vegetation phenology?

6. I suggest the authors add a paragraph of limitation discussion to clarify the limitation or uncertainty of data and methods in this study. For example, the uncertainty of remote sensing data including NDVI data (e.g., Ma et al., 2022; Shen et al., 2022) may affect the research results.

7. How did the authors consider differences in Phenology in different ecosystems (e.g., forests, grasslands, wetlands, etc.)?

8. In order to further highlight the innovation of this article, it is better to compare the results of this study with some related studies.

References:

Variation of vegetation autumn phenology and its climatic drivers in temperate grasslands of China. International Journal of Applied Earth Observation and Geoinformation, 2022, 114: 103064.

Vegetation greening, extended growing seasons, and temperature feedbacks in warming temperate grasslands of China. Journal of Climate, 2022, 35: 5103-5117.

6. PLOS authors have the option to publish the peer review history of their article (what does this mean? ). If published, this will include your full peer review and any attached files.

**Do you want your identity to be public for this peer review?** For information about this choice, including consent withdrawal, please see our Privacy Policy .

Reviewer #1: No

---

## [Author Response · Author response to Decision Letter 1]

15 Jul 2025

Dear editor and reviewers,

We hereby express our sincere gratitude to the editor and reviewers for their painstaking efforts in evaluating our manuscript entitled “Characteristics and driving mechanisms of vegetation phenology variations in the Bosten Lake Basin, Xinjiang, China” (ID: PLOS-D-25-17547R1). We are particularly grateful for your perceptive and constructive feedback, which has significantly improved the quality of our work. The manuscript has been meticulously re-examined, and all necessary modifications have been made in accordance with the reviewers' comments. Each comment has been systematically addressed, as elaborated in the attached revision description.

The revised version now includes references that did not appear in the original text. New references have been added to the manuscript, with revised reference numbers. The newly added references are displayed in red text. All of our revisions are highlighted in red text throughout the manuscript. We are honored to submit the revised version of our manuscript to PLOS ONE. The key revisions in the new manuscript are as follows:

1.Please ensure that your manuscript meets PLOS ONE's style requirements, including those for file naming. The PLOS ONE style templates can be found at https://journals.plos.org/plosone/s/file?id=wjVg/PLOSOne_formatting_sampl_main_body.pdf and https://journals.plos.org/plosone/s/file?id=ba62/PLOSOne_formatting_sample_title_authors_affiliations.pdf

Response: Thanks to the editors for their comments. We have revised the title and author attribution formatting in the manuscript according to the manuscript style template available at the following URL to bring it in line with the journal requirements.

2.In your Methods section, please provide additional information regarding the permits you obtained for the work. Please ensure you have included the full name of the authority that approved the field site access and, if no permits were required, a brief statement explaining why.

Response: We thank the editors for their comments and suggestions. No specific permits were required for this study because the fieldwork did not involve endangered/protected species, privately owned lands, or regulated activities. The study complied with all relevant local regulations.

3.Please note that PLOS ONE has specific guidelines on code sharing for submissions in which author-generated code underpins the findings in the manuscript. In these cases, we expect all author-generated code to be made available without restrictions upon publication of the work. Please review our guidelines at https://journals.plos.org/plosone/s/materials-and-software-sharing# loc-sharing-code and ensure that your code is shared in a way that follows best practice and facilitates reproducibility and reuse.

Response: We thank the editors for their comments and suggestions. The data generated by the authors to support the results of this study is stored in a public repository. The data can be accessed at: https://github.com/Likunyu7/dataC.

4.Thank you for stating in your Funding Statement: [This study was supported by the National Natural Science Foundation of China (Grant No. 42461042; 41561051) and the Natural Science Foundation of Xinjiang Uygur Autonomous Region, China (Grant No. 2023D01A44).].Please provide an amended statement that declares *all* the funding or sources of support (whether external or internal to your organization) received during this study, as detailed online in our guide for authors at http://journals.plos.org/plosone/s/submit-now.  Please also include the statement “There was no additional external funding received for this study.” in your updated Funding Statement. Please include your amended Funding Statement within your cover letter. We will change the online submission form on your behalf.

Response: This study was supported by the National Natural Science Foundation of China (Grant Nos. 42461042 and 41561051) and the Natural Science Foundation of Xinjiang Uygur Autonomous Region, China (Grant No. 2023D01A44). There was no additional external funding received for this study.

5.When completing the data availability statement of the submission form, you indicated that you will make your data available on acceptance. We strongly recommend all authors decide on a data sharing plan before acceptance, as the process can be lengthy and hold up publication timelines. Please note that, though access restrictions are acceptable now, your entire data will need to be made freely accessible if your manuscript is accepted for publication. This policy applies to all data except where public deposition would breach compliance with the protocol approved by your research ethics board. If you are unable to adhere to our open data policy, please kindly revise your statement to explain your reasoning and we will seek the editor's input on an exemption. Please be assured that, once you have provided your new statement, the assessment of your exemption will not hold up the peer review process.

Response: We thank the editors for their comments and suggestions. We make all data from this thesis unconditionally available via Zenodo upon publication, with an explanatory document to ensure reusability.

6.We note that Figures 1a-c, 5a,5c,5e, 7a-f in your submission contain [map/satellite] images which may be copyrighted.

Response: We are very grateful for the editors' opinions and suggestions. Among them, Figure 1a was created based on the standard map of the Ministry of Natural Resources of China (GS(2024)0650). No modifications were made to the boundaries of the base map, and this has been explained in the revised manuscript (Lines 120-122). The image boundaries in Figures 1b-c, 5a-f, and 7a-f are the image boundaries extracted by the authors themselves. These boundaries were extracted using the watershed extraction and analysis method based on the DEM data downloaded from the Google Earth Engine platform (https://earthengine.google.com/), and no data permission was required. All the pictures comply with the CC BY 4.0 license requirements of PLOS ONE.

Reviewer #1:

Major comments:

1.More mechanism explanations should be added to further explain the impact of climate change and human activities on vegetation phenology.

Response: We sincerely thank the reviewers for their constructive suggestions. We agree with the reviewers' comments and, with reference to relevant studies, have incorporated evapotranspiration and albedo as two factors to more comprehensively explore the driving mechanisms of vegetation phenological changes. These two factors have been added to both the correlation analysis and the PLS-PM analysis. Meanwhile, the relevant sections of the manuscript have been carefully revised and improved. For details, please refer to lines 137-145, 363-373, and 398-426 of the manuscript.

2.In the last paragraph of the introduction, the authors should highlight the significance of this study.

Response: We sincerely appreciate the reviewers' pertinent suggestions. We agree with the reviewers' opinions and have carefully reviewed and revised the manuscript in accordance with their suggestions. The paragraph of the introduction (Lines 37-50 and 78-100) has been rewritten to emphasize the significance and innovation of this study.

3. In Figure 1, the compass is not aligned with the direction of the latitude/longitude grid. I suggest the authors delete the compass and add the label of “China” in the Figure 1(a).

Response: We sincerely thank the reviewers for their constructive suggestions. We agree with the reviewers' opinions and have carefully reviewed and revised Figure 1 of the manuscript in accordance with their suggestions. To more accurately reflect the geographical location of the study area, we modified Figure 1(a) to "Xinjiang Region of China" and removed the compass, thereby improving the accuracy and professionalism of the map. In addition, we have marked the review number of the base map used to create this map to enhance its compliance and authority (Lines 119-122).

4. How did the authors account for inconsistent spatial resolution between different data sources?

Response: We sincerely thank the expert for this question and hereby reply to the relevant issues as follows. During the research process, different data sources need to be selected according to the research requirements, and the spatial resolutions of different data sources often vary. Usually, the spatial resolution of the main data is taken as the standard, and other data are resampled to unify the observation scale of all data. The spatial resolution of the MODIS NDVI data mainly used in this study is 250 m. Although the spatial resolutions of other auxiliary data differ from it, to facilitate the subsequent data calculation and analysis, we adopted the nearest neighbor resampling method to uniformly adjust their spatial resolutions to 250 m, which is consistent with that of the NDVI data. Meanwhile, we have provided detailed supplementary explanations in lines 141-144 of the revised manuscript.

5. How did authors consider the impact of other environmental variables (e.g., albedo, evapotranspiration, etc.) on vegetation phenology?

Response: We sincerely thank you for your valuable suggestions. In response to your suggestion, we have added two factors, evapotranspiration and albedo, to the manuscript for analyzing the driving mechanism and explored their influences on vegetation phenology. This is mainly reflected in the inclusion of evapotranspiration and albedo in the correlation analysis of phenology and driving factors (Figure 9), as well as the driving path analysis of phenological changes (Figure 10), with the aim of investigating their correlation with phenology and their direct and indirect impacts on vegetation phenological changes. The main modifications can be found in lines 363-373 and 398-424 of this manuscript. Additionally, in lines 477 to 488 of this manuscript., the discussion on the driving paths of influencing factors on vegetation phenology has been supplemented.

6. I suggest the authors add a paragraph of limitation discussion to clarify the limitation or uncertainty of data and methods in this study. For example, the uncertainty of remote sensing data including NDVI data (e.g., Ma et al., 2022; Shen et al., 2022) may affect the research results.

Response: We sincerely thank the reviewers for their constructive suggestions. We agree with the reviewers' opinions and have carefully revised and improved the manuscript in accordance with the experts' suggestions. In this manuscript, a paragraph has been added to discuss the limitations of the data and methods (lines 495-514). These include the uncertainty in calculating vegetation phenology using NDVI data from different sources, the potential impact that preprocessing remote sensing image data may have on the data, and an exploration of the uncertainty of how phenology extraction methods affect the results.

7. How did the authors consider differences in Phenology in different ecosystems (e.g., forests, grasslands, wetlands, etc.)?

Response: Thank you for your valuable suggestions. This study previously discussed the phenological differences and changing trends of different vegetation types, as illustrated in Figures 4 and 8. It covers seven vegetation types, namely Alpine Vegetation (AV), Meadows (Md), Marshes (Ms), Steppes (Sp), Needleleaf Forests (NF), Shrubland (Sl), and Cultivated Vegetation (CV). In this revision, specific analyses and descriptions have been added. For details, please refer to the text marked in red on lines 277-283.

8. In order to further highlight the innovation of this article, it is better to compare the results of this study with some related studies.

Response: We sincerely appreciate the reviewers for their constructive suggestions. We concur with the reviewers' comments and have meticulously reviewed and revised the manuscript in line with their inputs. Specifically, in lines 453-464 and 477-488 of this manuscript within the Discussion, the findings of this manuscript are compared with those of existing studies, with a deliberate focus on emphasizing the innovations of the present research.

We would like to express our sincere gratitude to the editors and reviewers for their dedicated efforts. We trust that the revisions made will meet with your approval, and we are truly thankful for your valuable comments and suggestions, which have been instrumental in enhancing the quality of our manuscript. On behalf of all co-authors, we appreciate your consideration and look forward to your reply.

Best regards.

Kunyu Li

E-mail: likunyu@stu.xjnu.edu.cn

Corresponding author: Xuemei Wang

E-mail: xmwang2022@xjnu.edu.cn

---

## [Decision Letter · Decision Letter 1]

7 Sep 2025

Characteristics and Driving Mechanisms of Vegetation Phenology Variations in the Bosten Lake Basin, Xinjiang, China

PONE-D-25-17547R1

Dear Dr. Wang,

We’re pleased to inform you that your manuscript has been judged scientifically suitable for publication and will be formally accepted for publication once it meets all outstanding technical requirements.

Kind regards,

Laxmi Kant Sharma, PhD

Academic Editor

PLOS ONE

Reviewers' comments:

Reviewer's Responses to Questions

**Comments to the Author**

1. If the authors have adequately addressed your comments raised in a previous round of review and you feel that this manuscript is now acceptable for publication, you may indicate that here to bypass the “Comments to the Author” section, enter your conflict of interest statement in the “Confidential to Editor” section, and submit your "Accept" recommendation.

Reviewer #1: All comments have been addressed

Reviewer #2: All comments have been addressed

2. Is the manuscript technically sound, and do the data support the conclusions?

Reviewer #1: Yes

Reviewer #2: Yes

3. Has the statistical analysis been performed appropriately and rigorously? 

Reviewer #1: Yes

Reviewer #2: Yes

4. Have the authors made all data underlying the findings in their manuscript fully available?

Reviewer #1: Yes

Reviewer #2: Yes

5. Is the manuscript presented in an intelligible fashion and written in standard English?

Reviewer #1: Yes

Reviewer #2: Yes

6. Review Comments to the Author

Reviewer #1: (No Response)

Reviewer #2: Recommendation for Acceptance

After a thorough review of the revised manuscript, now its find it to be scientifically sound, methodologically rigorous, and highly relevant to the journal’s scope. The authors have responded comprehensively to all editorial and reviewer comments, substantially improving the clarity, depth, and overall quality of the work.

The manuscript is now clearer, with improved figures, corrected map compliance (use of standard Chinese base maps with license compliance), and well-organized results showing both spatial and temporal patterns. Limitations and uncertainties—especially regarding NDVI data accuracy and method constraints—are explicitly acknowledged, aligning with best scientific practices.

The manuscript presents original, well-executed research with clear scientific contributions. The revisions have resolved all substantive concerns, and the study now meets the technical, ethical, and stylistic standards of PLOS ONE. I recommend acceptance without further revision.

7. PLOS authors have the option to publish the peer review history of their article (what does this mean? ). If published, this will include your full peer review and any attached files.

**Do you want your identity to be public for this peer review?** For information about this choice, including consent withdrawal, please see our Privacy Policy .

Reviewer #1: No

Reviewer #2: No

---

## [Editor Report · Acceptance letter]

PONE-D-25-17547R1

PLOS ONE

Dear Dr. Wang,

I'm pleased to inform you that your manuscript has been deemed suitable for publication in PLOS ONE. Congratulations! Your manuscript is now being handed over to our production team.

Kind regards,

on behalf of

Prof.Dr. Laxmi Kant Sharma

Academic Editor

PLOS ONE